# Non-Gaussian Pseudolinear Kalman Filtering-Based Target Motion Analysis with State Constraints

**Ming Li** [1], **Xiafei Tang** [2], **Qichun Zhang** [3] **and Yiqun Zou** [1,*]

1 School of Automation, Central South University, Changsha 410083, China
2 School of Electrical and Information Engineering, Changsha University of Science and Technology, Changsha 410114, China
3 Department of Computer Science, University of Bradford, Bradford BD7 1DP, UK
* Correspondence: yiqunzou@csu.edu.cn

**Abstract:** For the bearing-only target motion analysis (TMA), the pseudolinear Kalman filter (PLKF) solves the complex nonlinear estimation of the motion model parameters but suffers serious bias problems. The pseudolinear Kalman filter under the minimum mean square error framework (PL-MMSE) has a more accurate tracking ability and higher stability compared to the PLKF. Since the bearing signals are corrupted by non-Gaussian noise in practice, we reconstruct the PL-MMSE under Gaussian mixture noise. If some prior information, such as state constraints, is available, the performance of the PL-MMSE can be further improved by incorporating state constraints in the filtering process. In this paper, the mean square and estimation projection methods are used to incorporate PL-MMSE with linear constraints, respectively. Then, the linear approximation and second-order approximation methods are applied to merge PL-MMSE with nonlinear constraints, respectively. Simulation results demonstrate that the constrained PL-MMSE algorithms result in lower mean square errors and bias norms, which demonstrates the superiority of the constrained algorithms.

**Keywords:** bearing-only TMA; PLKF; PL-MMSE under gaussian mixture noise; linear and nonlinear constraints; constrained PL-MMSE algorithms

## 1. Introduction

Target motion analysis (TMA) refers to the real-time estimation of the position, speed, and other motion parameters of the tracked target by using sensors to obtain the measured information of the target by signal processing technology [1–3]. It has many applications in civilian and military fields, including military reconnaissance, intelligent transportation systems, and satellite navigation systems. The measurement information includes the angle of arrival (AOA) [4], time of arrival (TOA), time difference of arrival (TDOA) [5], and received signal strength (RSS) [6]. In this paper, we focus on AOA-based TMA, i.e., analyzing the target motion based on the bearing-only data emitted from the motion target and collected by the sensors.

The main difficulty of the bearing-only TMA is how to handle the nonlinear characteristic of the measurement equation. Methods for dealing with bearing-only problems can generally be divided into three categories. The first category is developed from the perspective of statistics [7]. The maximum likelihood estimator (MLE) uses the iterative optimization method to solve nonlinear equations to obtain the target position estimation. Since then, evolved methods [8,9] have been proposed to tackle the TMA problems. In [8], optimizing the likelihood function equipped with extra penalized terms gives the result that has a lower Cramér-rao bound than the standard estimator. The second category is the Kalman filter (KF) and its related methods. Due to the poor initialization, the standard Kalman filter [10] has shortcomings in robustness, convergence speed, and tracking accuracy. Many variant structures of the KF have been proposed to solve the nonlinear estimation problem. For example, Bucy et al. [11] proposes the nonlinear extended Kalman

filter (EKF). Julier et al. [12,13] proposes the method of the unscented Kalman filter. The particle filter [14–16] (PF) is also used for the bearing-only target motion analysis. Zheng Yi et al. [17] proposes an initial value optimization method for inverse smoothing filtering. This method effectively solves the problem that Kalman filtering methods are sensitive to initial value selection and reduces the estimation error. The third category is to linearize the nonlinear angle measurement equation by using the pseudolinear estimator (PLE) method [18]. The pseudolinear Kalman filter (PLKF) [19,20] is produced by combining the Kalman filter with the pseudolinear estimator method. Compared with other filtering methods, the main advantages of PLKF are high stability, good tracking performance, and small initial error under lower computational complexity [21]. However, the PLKF has a large bias due to the correlation between the measurement matrix and the pseudo-linear noise variable. Hence, several methods have been proposed to improve the performance of PLKF by compensating or reducing the pseudolinear estimation bias, including the modified pseudolinear estimator (MPLE) [22], bias-compensated PLKF (BC-PLKF) [23], IV Kalman filter (IVKF) [24] and IVKF based on the selective-angle-measurement (SAM-IVKF) [25] strategy. These variants of PLKF based on bias compensation are not always perfect when the measurement noise is large and the geometry is unfavorable. Based on the PLKF, Bu et al. [26] proposes a new pseudolinear filter under the minimum mean square error (PL-MMSE) framework without offset compensation, which shows better tracking performance under the large measurement noise than the above algorithms.

If the prior information, such as linear or nonlinear constraints on motion state, is available, we can take these conditions into consideration to improve the state estimation [27]. For example, tracking the vehicle driven on a straight or curved road is a constrained state estimation problem with the available road information [28]. Similar models also appear in other engineering applications, including the compartmental models method [29], turbofan engine health estimation [30] and so on. To estimate the states in such systems, some methods have been proposed, e.g., the model parameters reduction method [31], perfect measurements approach [32], estimation projection [33], linear approximation [33], second-order approximation [34,35]. For linear constraints, the model parameters' reduction method [31] transforms the constrained state estimation to the unconstrained state estimation. However, the reduction of the state constrained equations makes the interpretation such as the physical meaning of the states more difficult. The perfect measurements approach [32] adds state equality constraints into the measurement equation. The method increases the dimension of the state estimation problem and hence increases the computation effort. Estimation projection [33] incorporates the equality constraints into the state estimation frame. It projects the unconstrained state estimation to the constrained surface. For nonlinear constraints, linear approximation [33] uses the Taylor series expansion to the nonlinear state constraints. This method linearizes the nonlinear state constraints by keeping only the first-order terms. Distinct from the linear approximation, second-order approximation [34,35] keeps both the first-order and second-order terms to maintain the nonlinearity of constraints.

In practice, the bearing noise of the sensor is not always Gaussian. For example, the measurement disturbance is described by distribution with impulsive (heavy-tailed) properties in [36]. The performance of standard Kalman filters based on the MMSE framework does not behave well under such noise [37]. To study such heavy-tailed signals, Ref. [38] proposes a suitable method to approximate the heavy-tailed gamma distribution of random telegraph noise by Gaussian mixture distribution. Inspired by [38], the PL-MMSE is extended to estimate the bearings-only target motion model parameters in the presence of Gaussian mixture noise as the first contribution of this paper. This contribution can be deemed as the application of PL-MMSE under heavy-tailed noise with adaptive adjustment of the noise weights. Secondly, we focus on merging the PL-MMSE with constraints by four approaches to address TMA. The mean square method is applied with the PL-MMSE for linear constraints by minimizing the conditional mean square error subject to the state constraints. The estimation projection method is incorporated into the PL-MMSE by pro-

jecting the unconstrained estimate onto the constrained surface. For nonlinear constraints, the linear approximation method is to linearly approximate the nonlinear constraints by using Taylor series expansion. Then, the estimation projection method follows to equip with the PL-MMSE filter. The second-order approximation method views the nonlinear constraint function as a second-order approximation to the nonlinearity. It constructs an extra optimization step after the PL-MMSE by projecting an unconstrained state estimation onto a nonlinear constrained surface and solves this optimization to realize the estimation. Finally, the PL-MMSE filter with state constraints is tested for TMA on the straight line and the arc section. Experimental results confirm that the behavior of our constrained method is better than other competitors.

The rest of this paper is organized as follows. In Section 2, the PL-MMSE under Gaussian mixture noise is designed after the notations are introduced. Sections 3 and 4 combine constrained estimation technologies with the PL-MMSE to derive the PL-MMSE filter with linear and nonlinear state constraints, respectively. Section 5 simulates the two bearings-only TMA examples to show the sound performance of the constrained PL-MMSE algorithms. Section 6 concludes the whole paper and points out future research directions.

## 2. PL-MMSE Kalman Filter Under Gaussian Mixture Noise

In the bearing-only two-dimensional (2D) plane TMA, the target-sensor model is established, as shown in Figure 1.

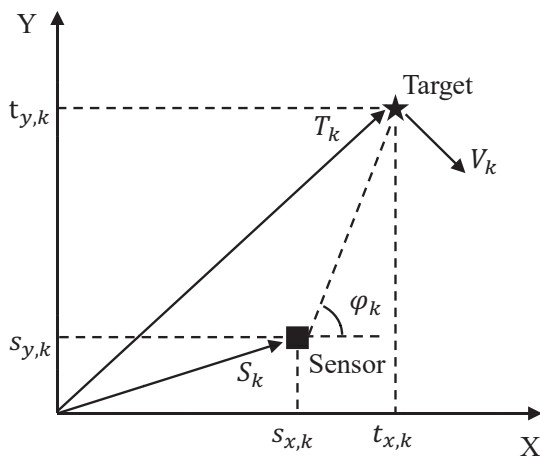

**Figure 1.** Schematic diagram of 2D bearing-only TMA.

As shown in Figure 1, the moving target position and velocity are $T_k = \begin{bmatrix} t_{x,k} \ t_{y,k} \end{bmatrix}^T$ and $V_k = \begin{bmatrix} v_{x,k} \ v_{y,k} \end{bmatrix}^T$, respectively, where

$$t_{x,k} = t_{x,k-1} + v_{x,k-1}T, \tag{1}$$
$$t_{y,k} = t_{y,k-1} + v_{y,k-1}T. \tag{2}$$

$T$ is the sampling interval. The sensor locates at $S_k = \begin{bmatrix} s_{x,k} \ s_{y,k} \end{bmatrix}^T$. The real angle information received from the sensor is given by

$$\varphi_k = \tan^{-1}\left(\frac{t_{y,k} - s_{y,k}}{t_{x,k} - s_{x,k}}\right). \tag{3}$$

The bearing measurement is

$$\hat{\varphi}_k = \varphi_k + e_k, \tag{4}$$

which indicates that the sensor measurement is corrupted by the mixed Gaussian noise $e_k$ with zero mean at time $kT$ $(k = 1, 2, 3, ..., n)$. The Gaussian mixture noise

$$e_k \sim \sum_{j=1}^{n} \lambda_j \mathcal{N}(0, \sigma_j^2) \tag{5}$$

is composed of $n$ independent distributed Gaussian noises with zero mean and variance $\sigma_j^2$, respectively, with

$$\lambda_j > 0 \quad \text{and} \quad \sum_{j=1}^{n} \lambda_j = 1. \tag{6}$$

The moving target state vector $X_k$ is given by

$$X_k = \begin{bmatrix} t_{x,k} & t_{y,k} & v_{x,k} & v_{y,k} \end{bmatrix}^T. \tag{7}$$

According to (3) and (4), the pseudolinear measurement equation between the target and the sensor is

$$\sin \hat{\varphi}_k t_{x,k} - \cos \hat{\varphi}_k t_{y,k} = \sin \hat{\varphi}_k s_{x,k} - \cos \hat{\varphi}_k s_{y,k} + \| T_k - S_k \| \sin e_k. \tag{8}$$

The pseudolinear state-space model for bearings-only TMA is

$$X_k = F X_{k-1} + \omega_{k-1}, \tag{9}$$
$$Z_k = H_k X_k + \tau_k, \tag{10}$$

where $X_k$ and $X_{k-1}$ are the motion states at time $kT$ and $(k-1)T$, respectively. It is assumed that $\omega_{k-1}$ is the Gaussian mixture noise composed of n independent distributed Gaussian noises with zero mean and variance $\xi_i^2$, respectively. According to the motion model (8), the state transition matrix F and the process noise Jacobian D are

$$F = \begin{bmatrix} 1 & 0 & T & 0 \\ 0 & 1 & 0 & T \\ 0 & 0 & 1 & 0 \\ 0 & 0 & 0 & 1 \end{bmatrix}, \tag{11}$$

$$D = \begin{bmatrix} 0 & 0 & T^2/2 & 0 \\ 0 & 0 & 0 & T^2/2 \\ 0 & 0 & T & 0 \\ 0 & 0 & 0 & T \end{bmatrix}. \tag{12}$$

Ref. [39]. In the PLKF algorithm, the estimated state of the target at time $kT$ is

$$\hat{X}_{k|k-1} = F \hat{X}_{k-1}. \tag{13}$$

The updated covariance matrix is

$$P_{k|k-1} = F P_{k-1|k-1} F^T + D Q_{k-1} D^T \tag{14}$$

where the Gaussian mixture noise variance $Q_{k-1}$ is

$$Q_{k-1} = \sum_{i=1}^{n} \rho_i \xi_i^2 \tag{15}$$

with

$$\rho_j > 0 \quad \text{and} \quad \sum_{i=1}^{n} \rho_i = 1. \tag{16}$$

The observation matrix is

$$H_k = [\sin \hat{\varphi}_i \ -\cos \hat{\varphi}_i \ 0 \ 0]^T. \tag{17}$$

The Kalman gain is

$$K_k = P_{k|k-1} H_k^T \left[ H_k P_{k|k-1} H_k^T + R_k \right]^{-1} \tag{18}$$

where the pseudolinear noise variance $R_k$ is given by

$$R_k = \|T_k - S_k\| \left( \sum_{i=1}^{n} \lambda_i e^{\frac{1-e^{\frac{-2\sigma_i^2}{2}}}{2}} \right). \tag{19}$$

The updated pseudolinear measurement is

$$\hat{Z}_{k|k-1} = H_k \hat{X}_{k|k-1}. \tag{20}$$

Therefore, the target state and covariance update equation at time $kT$ can be described by

$$\hat{X}_{k|k} = \hat{X}_{k|k-1} + K_k(Z_k - \hat{Z}_{k|k-1}), \tag{21}$$

$$P_{k|k} = P_{k|k-1} - K_k H_k P_{k|k-1}. \tag{22}$$

Based on the PLKF, the PL-MMSE for Gaussian mixture noise can be rewritten as shown in Table 1 where the pseudolinear observation matrix is given by

$$H_k^1 = [\tan^{-1}(\hat{X}_{k|k-1}(2) - S_k(2)) \ \tan^{-1}(\hat{X}_{k|k-1}(1) - S_k(1)) \ 0 \ 0]. \tag{23}$$

**Table 1.** PL-MMSE under Gaussian mixture noise.

| |
|---|
| 1. Initialization<br>$\hat{x}_0 = \bar{x}_0$<br>$P_0 = E\left[(x_0 - \bar{x}_0)(x_0 - \bar{x}_0)^T\right]$ |
| 2. State Prediction<br>$\hat{x}_{k|k-1} = F_{k-1}\hat{x}_{k-1}$ |
| 3. Covariance Prediction<br>$P_{k|k-1} = F_{k-1}P_{k-1|k-1}F_{k-1}^T + DQ_kD^T$ |
| 4. Filter Gain<br>$K_1 = \sum_{j=1}^{n} \lambda_j e^{\frac{-\sigma_j^2}{2}} P_{k|k-1}\left(H_k^1\right)^T$<br>$K_2 = H_k P_{k|k-1} H_k^T + \sum_{i=1}^{n} \rho_i e^{\frac{e^{-2\zeta_{i-1}^2}}{2}} \left[P_{k|k-1}(1,1) + P_{k|k-1}(2,2)\right] + R_k$<br>$K_k = K_1 K_2^{-1}$ |
| 5. State Update<br>$\hat{x}_k = \hat{x}_{k|k-1} + K_k(z_k - H_k\hat{x}_{k|k-1})$ |
| 6. Covariance Update<br>$P_{k|k} = P_{k|k-1} - K_k K_1^T$ |
| 7. $k = k + 1$, go to 2. |

## 3. PL-MMSE Kalman Filter with Linear State Constraints

Assume that the motion state is bounded by linear constraints

$$Gx_k = g \tag{24}$$

where $G$ is a known $R^{d \times n}$ matrix while $g$ is a known $R^{d \times 1}$ vector with $d < n$. It is also assumed that $G$ has full rank. Next, we introduce two methods to encapsulate the PL-MMSE with linear state constraints.

### 3.1. Mean Square Method

The idea of the mean square method is to obtain the state estimation $\tilde{x}$ of the moving target with linear constraints by minimizing the conditional mean square error. Let

$$\tilde{x}_k = \min_{\tilde{x}_k} E(\|x_k - \tilde{x}_k\|^2 | Z_k) \quad s.t. \quad G\tilde{x}_k = g \tag{25}$$

where

$$
\begin{aligned}
E(\|x - \tilde{x}\|^2 | Z) &= \int (x - \tilde{x})^T (x - \tilde{x}) P(x|Z) dx \\
&= \int x^T x P(x|Z) dx - 2\tilde{x}^T \int x P(x|Z) dx + \tilde{x}^T \tilde{x}.
\end{aligned} \tag{26}
$$

A Lagrangian function is constructed to solve the constrained problem as

$$
\begin{aligned}
J &= E(\|x - \tilde{x}\|^2 | Z) + 2\lambda^T (G\tilde{x} - g) \\
&= \int x^T x P(x|Z) dx - 2\tilde{x}^T \int x P(x|Z) dx \\
&\quad + \tilde{x}^T \tilde{x} + 2\lambda^T (G\tilde{x} - g).
\end{aligned} \tag{27}
$$

The conditional mean of $x$ is

$$\hat{x} = \int x P(x|Z) dx. \tag{28}$$

After substituting (28) into (27), taking the partial derivatives of $\tilde{x}$ and $\lambda$, respectively, leads to

$$
\begin{aligned}
\frac{\partial J}{\partial \tilde{x}} &= -2\hat{x} + 2\tilde{x} + 2G^T \lambda \\
&= 0, \tag{29} \\
\frac{\partial J}{\partial \lambda} &= G\tilde{x} - g \\
&= 0. \tag{30}
\end{aligned}
$$

Solving (29) and (30) gives

$$\tilde{x} = \hat{x} - G^T (GG^T)^{-1} (G\hat{x} - g), \tag{31}$$

$$\lambda = (GG^T)^{-1} (G\hat{x} - g). \tag{32}$$

From (31), the constrained estimate of motion state is the unconstrained estimate minus the correction term.

### 3.2. Estimation Projection Method

As a standard method to deal with constraints, the estimation projection method obtains the constrained estimate $\tilde{x}$ by projecting the unconstrained estimate $\hat{x}$ onto the constrained surface. Define

$$\tilde{x} = arg \min_x (x - \hat{x})^T W (x - \hat{x}) \quad s.t. \quad Gx = g \tag{33}$$

where $W$ is a positive definite weighting matrix. The Lagrangian function used to solve this problem is

$$J = (x - \hat{x})^T W (x - \hat{x}) + 2\lambda^T (Gx - g). \tag{34}$$

The necessary conditions for the local minimum are given by

$$\frac{\partial J}{\partial x} = 0, \tag{35}$$

$$\frac{\partial J}{\partial \lambda} = 0. \tag{36}$$

Solving (35) and (36) gives

$$\tilde{x} = \hat{x} - W^{-1}G^T(GW^{-1}G^T)^{-1}(G\hat{x} - g), \tag{37}$$

$$\lambda = (GW^{-1}G^T)^{-1}(G\hat{x} - g). \tag{38}$$

It is worthwhile to point out that the result given by the estimation projection is equal to the mean square method when $W = I$. Hereby, Table 2 summarizes the steps of the PL-MMSE with linear constraints by the estimation projection method.

**Table 2.** PL-MMSE with Linear Constraints Algorithm.

| |
|---|
| 1. Initialization<br>$\hat{x}_0 = \bar{x}_0$<br>$P_0 = E[(x_0 - \bar{x}_0)(x_0 - \bar{x}_0)^T]$ |
| 2. Predict<br>$\hat{x}_{k\|k-1} = F_{k-1}\hat{x}_{k-1}$<br>$P_{k\|k-1} = F_{k-1}P_{k-1\|k-1}F_{k-1}^T + DQ_kD^T$ |
| 3. Filter Gain<br>$K_1 = \sum_{j=1}^n \lambda_j e^{\frac{-\sigma_i^2}{2}} P_{k\|k-1}(H_k^1)^T$<br>$K_2 = H_k P_{k\|k-1} H_k^T + \sum_{i=1}^n \rho_i e^{\frac{e^{-2\xi_{i-1}^2}}{2}}[P_{k\|k-1}(1,1) + P_{k\|k-1}(2,2)] + R_k$<br>$K_k = K_1 K_2^{-1}$ |
| 4. Update<br>$\hat{x}_k = \hat{x}_{k\|k-1} + K_k(z_k - H_k\hat{x}_{k\|k-1})$<br>$P_{k\|k} = P_{k\|k-1} - K_k K_1^T$ |
| 5. When the linear constraint is $Gx_k = g$,<br>$\tilde{x}_k = \hat{x}_{k\|k} - W^{-1}G^T(GW^{-1}G^T)^{-1}(G\hat{x}_{k\|k} - g)$ |
| 6. $k = k + 1$, go to 2. |

## 4. PL-MMSE Kalman Filter with Nonlinear State Constraints

Consider the nonlinear constraint on the system state is as

$$h(x) = q \tag{39}$$

where $h(\cdot)$ is a nonlinear function. $q$ is a scalar. Next, we address the nonlinear constraint using the linear approximation method and second-order approximation, respectively.

### 4.1. Linear Approximation

Use the Taylor series to expand (39) at $\hat{x}$ as

$$h(x) - q = h(\hat{x}) + h'(\hat{x})^T(x - \hat{x} + \frac{1}{2!}(x - \hat{x})^T h''(\hat{x})(x - \hat{x}) + \cdots - q$$

$$= 0 \tag{40}$$

where $h^{'}(\cdot)$ denotes the Jacobian matrix of $h(\cdot)$ and $h^{''}(\cdot)$ is the Hessian matrix of $h(\cdot)$. Using only the first-order term to approximate the nonlinear state constraint leads to

$$h^{'}(\hat{x})^T x \approx q - h(\hat{x}) + h^{'}(\hat{x})^T \hat{x}. \tag{41}$$

Through observation, (41) has a similar structure with (24) where $G$ of (24) is replaced with $h^{'}(\hat{x})^T$ in (41) and $g$ with $g - h(\hat{x}) + h^{'}(\hat{x})^T \hat{x}$. After applying the estimation projection method, the constrained estimator for the linear approximation method becomes

$$\tilde{x}_k = \hat{x}_{k|k} - (h'(\hat{x}_{k|k}))^T (h'(\hat{x}_{k|k})(h'(\hat{x}_{k|k}))^T)^{-1}(h(\hat{x}_{k|k}) - g). \tag{42}$$

*4.2. Second-Order Approximation*

When the first and second-order terms are both kept, (40) can be rewritten into

$$\begin{aligned} f(x) &= \begin{bmatrix} x^T & 1 \end{bmatrix} \begin{bmatrix} M & m \\ m^T & m_0 \end{bmatrix} \begin{bmatrix} x \\ 1 \end{bmatrix} \\ &= x^T M x + 2m^T x + m_0 \\ &= 0. \end{aligned} \tag{43}$$

Here

$$M = \frac{1}{2} h''(\hat{x}_{k|k}), \tag{44}$$

$$m = (h'(\hat{x}_{k|k}) - \hat{x}_{k|k}^T h''(\hat{x}_{k|k}))^T / 2, \tag{45}$$

$$m_0 = h(\hat{x}_{k|k}) - h'(\hat{x}_{k|k})\hat{x}_{k|k} + (\hat{x}_{k|k})^T M \hat{x}_{k|k} - q. \tag{46}$$

Construct an optimization problem by projecting an unconstrained state estimation onto a nonlinear constrained surface, i.e.,

$$\tilde{x} = arg \min_{x}(z - Hx)^T(z - Hx) \quad s.t. \quad f(x) = 0 \tag{47}$$

The Lagrangian function is formed with the multiplier $\lambda$ as

$$J = (z - Hx)^T(z - Hx) + \lambda f(x). \tag{48}$$

The optimal solution can be found by solving

$$\begin{aligned} \frac{\partial J}{\partial x} &= -H^T z + \lambda m + (H^T H + \lambda M)x \\ &= 0, \end{aligned} \tag{49}$$

$$\begin{aligned} \frac{\partial J}{\partial \lambda} &= x^T M x + m^T x + x^T m + m_0 \\ &= 0. \end{aligned} \tag{50}$$

Assume the matrix $H^T H + \lambda M$ is invertible. The constrained solution $\tilde{x}$ can be expressed by

$$\tilde{x} = (H^T H + \lambda M)^{-1}(H^T z - \lambda m) \tag{51}$$

which is the unconstrained solution when $\lambda = 0$.

Applying the Cholesky factorization to $M$ and $S = H^T H$ gives

$$M = L^T L, \tag{52}$$

$$S = E^T E \tag{53}$$

where $E$ is an upper right diagonal matrix. We can apply singular value decomposition (SVD) to the matrix $LE^{-1}$ as

$$LE^{-1} = U\Sigma V^T \tag{54}$$

where $U$ and $V$ are orthonormal matrices, and $\Sigma$ is a diagonal matrix with its diagonal elements denoted by $p_i$. In order to simplify (51), two additional vectors are defined as

$$
\begin{aligned}
e(\lambda) &= [\cdots \; e_i(\lambda) \; \cdots]^T \\
&= V^T(E^T)^{-1}(H^Tz - \lambda m), \tag{55}
\end{aligned}
$$

$$
\begin{aligned}
t &= [\cdots \; t_i \; \cdots]^T \\
&= V^T(E^T)^{-1}m. \tag{56}
\end{aligned}
$$

With these new matrices and vector notations, (51) can be expressed as

$$\tilde{x} = E^{-1}V(I + \lambda\Sigma^T\Sigma)^{-1}e(\lambda). \tag{57}$$

With (55), (56) and (57),

$$
\begin{aligned}
\tilde{x}^T M \tilde{x} &= e(\lambda)^T(I + \lambda\Sigma^T\Sigma)^{-T}\Sigma^T\Sigma(I + \lambda\Sigma^T\Sigma)^{-1}e(\lambda) \\
&= \sum_i \frac{e_i^2(\lambda)p_i^2}{(1 + \lambda p_i^2)^2}, \tag{58}
\end{aligned}
$$

$$
\begin{aligned}
m^T \tilde{x} &= t^T(I + \lambda\Sigma^T\Sigma)^{-1}e(\lambda) \\
&= \sum_i \frac{e_i(\lambda)t_i}{1 + \lambda p_i^2}, \tag{59}
\end{aligned}
$$

After plugging in (58) and (59), $f(x)$ transforms into

$$
\begin{aligned}
f(\lambda) &= e(\lambda)^T(I + \lambda\Sigma^T\Sigma)^{-T}\Sigma^T\Sigma(I + \lambda\Sigma^T\Sigma)^{-1}e(\lambda) \\
&\quad + t^T(I + \lambda\Sigma^T\Sigma)^{-1}e(\lambda) \\
&\quad + e(\lambda)^T(I + \lambda\Sigma^T\Sigma)^{-1}t + m_0 \\
&= \sum_i \frac{e_i^2(\lambda)p_i^2}{(1 + \lambda p_i^2)^2} + 2\sum_i \frac{e_i(\lambda)t_i}{1 + \lambda p_i^2} + m_0 \tag{60}
\end{aligned}
$$

Since (60) is a nonlinear equation of $\lambda$, it is difficult to obtain a closed-form solution. Numerical root-finding algorithms are used such as Newton method [40] to solve (60). The derivatives of $f(\lambda)$ and $e(\lambda)$ are

$$
\begin{aligned}
\dot{f}(\lambda) &= 2\sum_i \frac{e_i(\lambda)\dot{e}_i(1 + \lambda p_i^2)p_i^2 - e_i^2(\lambda)p_i^4}{(1 + \lambda p_i^2)^3} \\
&\quad + 2\sum_i \frac{\dot{e}_i t_i(1 + \lambda p_i^2) - e_i(\lambda)t_i\sigma_i^2}{(1 + \lambda p_i^2)^2}, \tag{61}
\end{aligned}
$$

$$
\begin{aligned}
\dot{e} &= [\ldots\dot{e}_i\ldots]^T \\
&= -V^T(G^T)^{-1}m. \tag{62}
\end{aligned}
$$

with respect to $\lambda$. Then, the iterative solution of $\lambda$ with Newton method can be given by

$$\lambda_{k+1} = \lambda_k - \frac{f(\lambda_k)}{\dot{f}(\lambda_k)}. \tag{63}$$

(63) starts with $\lambda_0 = 0$. If $|\lambda_{k+1} - \lambda_k| < \tau$ where $\tau$ is the tolerance, or the number of iterations reaches a preset value, the iteration stops. Then, we can obtain the constrained

state estimate of the moving target by substituting the solution of $\lambda$ into (57). Table 3 shows the steps of the PL-MMSE with nonlinear constraints by second-order approximation.

**Remark 1.** *The algorithm presented in Table 1 have many potential applications, which use the model shown in (9) and (10). For example, in the ocean environment, a self-moving ship monitors noisy sonar bearings to an acoustic target ship and then pours the measurements into the filters to estimate and predict the source position and velocity [19]. If the waterway of the target is known in advance, the constraint can be brought into the methods in Tables 2 and 3 to further raise the estimate accuracy.*

**Table 3.** PL-MMSE with Nonlinear Constraints Algorithm.

| |
|---|
| 1. Initialization<br>$\hat{x}_0 = \bar{x}_0$<br>$P_0 = E\left[(x_0 - \bar{x}_0)(x_0 - \bar{x}_0)^T\right]$ |
| 2. Predict<br>$\hat{x}_{k\|k-1} = F_{k-1}\hat{x}_{k-1}$<br>$P_{k\|k-1} = F_{k-1}P_{k-1\|k-1}F_{k-1}^T + DQ_kD^T$ |
| 3. Filter Gain<br>$K_1 = \sum_{j=1}^{n} \lambda_j e^{\frac{-\sigma_i^2}{2}} P_{k\|k-1}\left(H_k^1\right)^T$<br>$K_2 = H_k P_{k\|k-1} H_k^T + \sum_{i=1}^{n} \rho_i e^{\frac{e^{-2\xi_{i-1}^2}}{2}}\left[P_{k\|k-1}(1,1) + P_{k\|k-1}(2,2)\right] + R_k$<br>$K_k = K_1 K_2^{-1}$ |
| 4. Update<br>$\hat{x}_k = \hat{x}_{k\|k-1} + K_k(z_k - H_k\hat{x}_{k\|k-1})$<br>$P_{k\|k} = P_{k\|k-1} - K_k K_1^T$ |
| 5. When the nonlinear constraint is $x_k^T M x_k + m^T x_k + x_k^T m + m_0 = 0$<br>Use the iterative method to find $\lambda$<br>Then $\tilde{x}_k = G^{-1}V(I + \lambda_k \Sigma^T \Sigma)^{-1}e(\lambda_k)$ |
| 6. $k = k + 1$, go to 2. |

## 5. Simulation

This section simulates examples and compares the performance of PL-MMSE, PLKF, BC-PLKF, IVKF and the corresponding algorithms with state constraints for moving targets under linear or nonlinear constraints with Gaussian mixture noise. To clarify, the combination algorithm of PL-MMSE and the mean square method is defined as PL-MMSE-C ($W = I$). Similarly, PL-MMSE combined with the estimation projection method is set to PL-MMSE-C ($W = P^{-1}$). PL-MMSE incorporated with the linear approximation and second-order approximation methods are defined as PL-MMSE-L and PL-MMSE-S, respectively. Other constrained algorithms are named in the same way as above. Each simulation result is generated in $M_0 = 1000$ Monte Carlo experiments with $N = 200$ sampling time scans for each run.

### 5.1. Performance Metrics

As defined in this subsection, the performance is evaluated using the root mean square errors (RMSEs) and the bias norms (BNorms). The RMSE and BNorm of the target position estimation are

$$\text{RMSE}_k^{pos} = \sqrt{\frac{1}{M_0}\sum_{i=1}^{M_0}\left\|\hat{x}_{k|k}^i(1:2) - x_k^i(1:2)\right\|^2}, \tag{64}$$

$$\text{BNorm}_k^{pos} = \left\|\frac{1}{M_0}\sum_{i=1}^{M_0}\left(\hat{x}_{k|k}^i(1:2) - x_k^i(1:2)\right)\right\| \tag{65}$$

where $\hat{x}^i_{k|k}(1:2)$ is the estimated target position and $x^i_k(1:2)$ is the true target position at time $kT$ at the $i$th run, respectively.

The RMSE and BNorm of the target velocity estimation are

$$\text{RMSE}^{vel}_k = \sqrt{\frac{1}{M_0} \sum_{i=1}^{M_0} \|\hat{x}^i_{k|k}(3:4) - x^i_k(3:4)\|^2}, \tag{66}$$

$$\text{BNorm}^{vel}_k = \|\frac{1}{M_0} \sum_{i=1}^{M_0} \left( \hat{x}^i_{k|k}(3:4) - x^i_k(3:4) \right) \| \tag{67}$$

where $\hat{x}^i_{k|k}(3:4)$ is the estimated target velocity and $x^i_k(3:4)$ is the true target velocity at time $kT$ at the $i$th run.

Similarly, the time-averaged RMSE, BNorm of the target position and velocity estimation are

$$\text{RMSE}^{pos}_{avg} = \sqrt{\frac{1}{M_0 B} \sum_{i=1}^{M_0} \sum_{k=L_0}^{N} \|\hat{x}^i_{k|k}(1:2) - x^i_k(1:2)\|^2}, \tag{68}$$

$$\text{BNorm}^{pos}_{avg} = \frac{1}{B} \sum_{k=L_0}^{N} \|\frac{1}{M_0} \sum_{i=1}^{M_0} \left( \hat{x}^i_{k|k}(1:2) - x^i_k(1:2) \right) \|, \tag{69}$$

$$\text{RMSE}^{vel}_{avg} = \sqrt{\frac{1}{M_0 B} \sum_{i=1}^{M_0} \sum_{k=L_0}^{N} \|\hat{x}^i_{k|k}(3:4) - x^i_k(3:4)\|^2}, \tag{70}$$

$$\text{BNorm}^{vel}_{avg} = \frac{1}{B} \sum_{k=L_0}^{N} \|\frac{1}{M_0} \sum_{i=1}^{M_0} \left( \hat{x}^i_{k|k}(3:4) - x^i_k(3:4) \right) \|. \tag{71}$$

Here, we set $B = N - L_0 + 1$ with $L_0 = 50$ where $L_0$ is an offset parameter to reduce the time-averaged metrics affected by the initial tracking errors in the simulations.

### 5.2. Simulation Parameters

In order to objectively compare the performance of the constrained PL-MMSE with other algorithms, we adopt the same sensor moving trajectory as in [24], as shown in Figure 2. The sensor trajectory is divided into five constant velocity segments, where the end position of each segment trajectory is set as $[60\ 0]^T$ m, $[0\ 7.5]^T$ m, $[60\ 15]^T$ m, $[0\ 22.5]^T$ m, $[60\ 30]^T$ m and $[0\ 77.5]^T$ m. Starting from the initial position $r_0 = [60\ 0]^T$ m, the sensor takes the direct measurement value at every sampling interval $T = 0.1$ s. The bearing noise and process noise are a Gaussian mixture noise with zero mean. The estimated initial state $\hat{x}_{1|1}$ is sampled from the true initial state $x_1$ with a Gaussian mixture distribution, which has the initial covariance $P_{1|1}$.

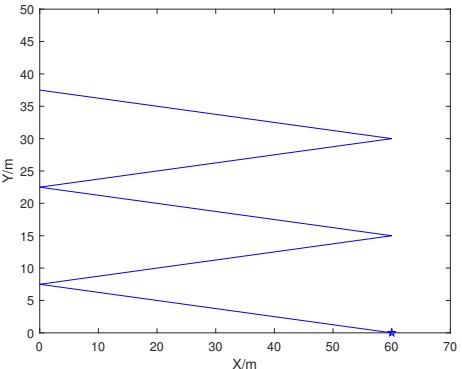

**Figure 2.** Sensor trajectory with five constant velocity segments and the initial position marked by a star.

### 5.3. Simulation Scenarios

Designed algorithms are tested in two scenarios in this subsection. As can be observed in Figure 3a, the target moves on a straight line in the first scenario while the target moves on an arc in the second scenario, as shown in Figure 3b with nearly constant velocity magnitude $V$.

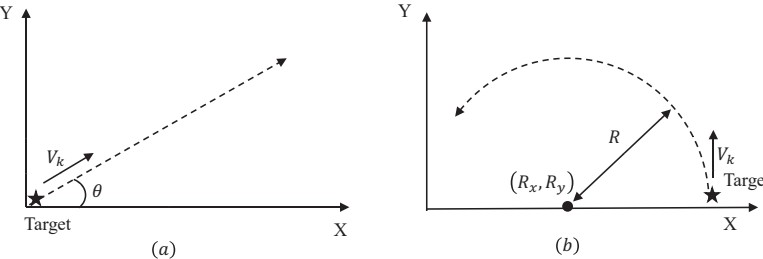

**Figure 3.** (**a**) target travels in a straight line along the direction $\theta$; (**b**) target travels along the circular road with the turning center $(R_x, R_y)$ and the radius $R$.

Under the known angle of the vehicle, the constrained matrix $G$ and the vector $g$ are governed by

$$G = \begin{bmatrix} 1 & -\tan\theta & 0 & 0 \\ 0 & 0 & 1 & -\tan\theta \end{bmatrix}, \tag{72}$$

$$g = [0, 0]^T. \tag{73}$$

The constrained estimate can be produced by setting $W = I$ and $W = P^{-1}$. In the simulation, the sampling interval T is set to 0.1 s. The total time span is the 20 s. The angle $\theta$ is set to $\pi/4$. The velocity $V$ is set to 12 m/s. The target initial position is $[0\ 0]^T$ m. The true initial state is set to $x_1 = \begin{bmatrix} 0 & 0 & 6\sqrt{2} & 6\sqrt{2} \end{bmatrix}^T$. The initial covariance matrix is $P_{1|1} = \text{diag}([1\ 1\ 0.01\ 0.01])$. In addition, the covariances of process noise $\omega_{k-1}$ and bearing noise $e_k$ are given by

$$\omega_{k-1} = \lambda\mathcal{N}(\mu_{x1}, Q_1) + (1-\lambda)\mathcal{N}(\mu_{x2}, Q_2), \tag{74}$$

$$e_k = \rho^2[\lambda\mathcal{N}(\mu_{z1}, R_1) + (1-\lambda)\mathcal{N}(\mu_{z2}, R_2)], \tag{75}$$

respectively, where $\mu_{x1}^T = \begin{bmatrix} 0 & 0 & 0 & 0 \end{bmatrix}$, $\mu_{x2}^T = \begin{bmatrix} 0 & 0 & 0 & 0 \end{bmatrix}$, $\mu_{z1}^T = 0$, $\mu_{z2}^T = 0$, $Q_1 = \text{diag}([0\ 0\ 0.15\ 0.15])$, $Q_2 = \text{diag}([0\ 0\ 0.23\ 0.23])$, $R_1 = 0.015$, $R_2 = 0.019$ and $\lambda = 0.4$. The variable $\rho$ controls the magnitude of bearing noise $e_k$. By setting $\rho$ from 1 to 10 with the difference of 1, Table 4 shows the standard deviation $\sigma_\theta$ of bearing noise for the corresponding $\rho$.

**Table 4.** Standard deviation $\sigma_\theta$ against $\rho$ in the first scenario.

| $\rho$ | 1 | 2 | 3 | 4 | 5 | 6 | 7 | 8 | 9 | 10 |
|---|---|---|---|---|---|---|---|---|---|---|
| $\sigma_\theta$ (°) | 1 | 2 | 3 | 4 | 5 | 6 | 7 | 8 | 9 | 10 |

Simulation results of the mean RMSEs and BNorms of the target position and velocity estimates against $\sigma_\theta$ are presented in Figure 4.

It is noticeable that the performance metric values of all algorithms increase and finally tend to be stable with $\sigma_\theta$ in Figure 4, where the performance of PL-MMSE is always better than that of PLKF, BC-PLKF, and IVKF. The $\text{RMSE}_{avg}^{pos}$ of PL-MMSE is stable at 2.5 m at a large bearing noise level, which significantly outperforms other unconstrained algorithms. The evolution of RMSEs, BNorms of the target position and velocity estimates in time $kT$ ($k = 1, \ldots 150$) for $\sigma_\theta = 7°$ is shown in Figure 5.

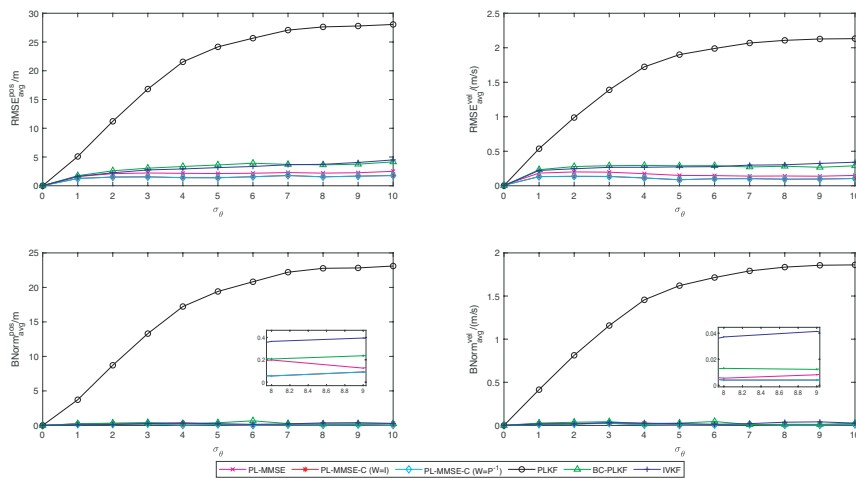

**Figure 4.** Time-averaged RMSEs, BNorms and bearing noise standard deviation for the PL-MMSE, PLKF, BC-PLKF and IVKF algorithms, as well as, the PL-MMSE algorithm with linear constraints proposed in the paper.

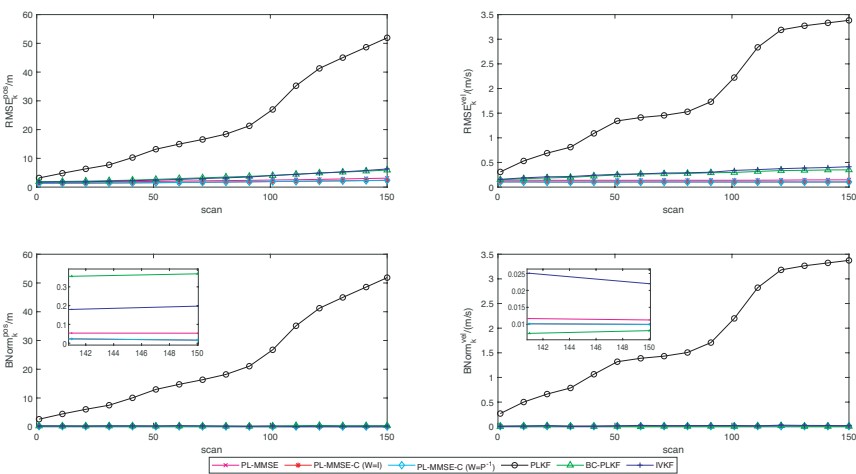

**Figure 5.** RMSEs, BNorms at different time scan for $\sigma_\theta = 7°$ for the PL-MMSE, PLKF, BC-PLKF and IVKF algorithms, as well as, the proposed PL-MMSE algorithm with linear constraints in the paper.

The tracking performance of all algorithms gradually deteriorates with the rising scan under a large bearing noise level. The metric values of PL-MMSE gradually approach $\text{RMSE}_k^{pos} = 3.11$ m, $\text{RMSE}_k^{vel} = 0.147$ m/s, $\text{BNorm}_k^{pos} = 0.053$ m, and $\text{BNorm}_k^{vel} = 0.011$ m/s, which are remarkably lower than other unconstrained algorithms. It can also be observed that the constrained PL-MMSE is superior to the unconstrained PL-MMSE at all bearing noise levels in Figures 4 and 5, which indicates the constrained algorithm has a better robustness and tracking performance. Comparisons of four algorithms combined with the mean square method and the estimation projection method for $\sigma_\theta = 7°$ are presented in Figures 6 and 7, respectively, which demonstrate that PL-MMSE-C ($W = I$) and PL-MMSE-C ($W = P^{-1}$) have less errors than other corresponding constrained algorithms at a large bearing noise level.

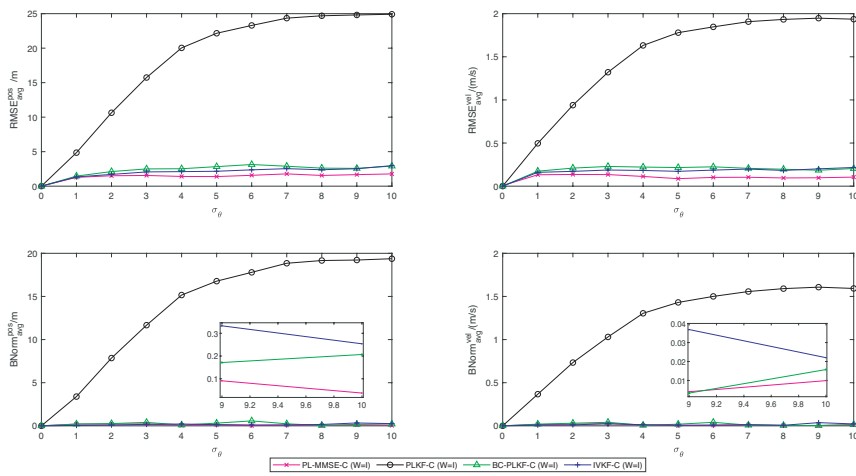

**Figure 6.** Time-averaged RMSEs, BNorms and bearing noise standard deviation for the four algorithms combined with the mean square method.

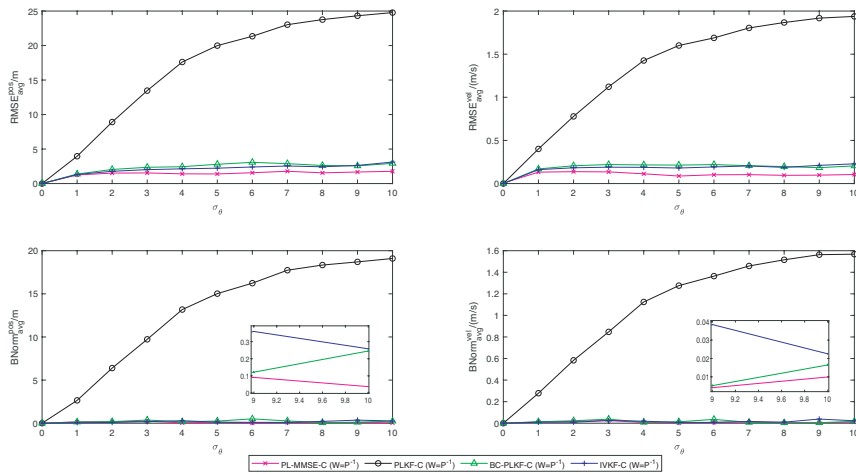

**Figure 7.** Time-averaged RMSEs, BNorms and bearing noise standard deviation for the four algorithms combined with the estimation projection method.

Table 5 shows the mean RMSEs and BNorms of different filters for $\sigma_\theta = 7°$.

**Table 5.** $RMSE_{avg}^{pos}$, $RMSE_{avg}^{vel}$, $BNorm_{avg}^{pos}$ and $BNorm_{avg}^{vel}$ of different filters for $\sigma_\theta = 7°$ at $V = 12$ m/s on the straight line.

| Filter | $RMSE_{avg}^{pos}$ | $RMSE_{avg}^{vel}$ | $BNorm_{avg}^{pos}$ | $BNorm_{avg}^{vel}$ |
|---|---|---|---|---|
| PL-MMSE | 2.294 | 0.140 | 0.139 | 0.011 |
| PLKF | 27.068 | 2.069 | 22.202 | 1.791 |
| BC-PLKF | 3.714 | 0.273 | 0.270 | 0.010 |
| IVKF | 3.648 | 0.299 | 0.219 | 0.021 |
| PL-MMSE-C ($W = I$) | 1.781 | 0.103 | 0.084 | 0.001 |
| PL-MMSE-C ($W = P^{-1}$) | 1.781 | 0.103 | 0.084 | 0.001 |
| PLKF-C ($W = I$) | 24.350 | 1.907 | 18.847 | 1.557 |
| PLKF-C ($W = P^{-1}$) | 23.027 | 1.804 | 17.739 | 1.458 |
| BC-PLKF-C ($W = I$) | 2.890 | 0.207 | 0.234 | 0.008 |
| BC-PLKF-C ($W = P^{-1}$) | 2.883 | 0.207 | 0.267 | 0.008 |
| IVKF-C ($W = I$) | 2.537 | 0.198 | 0.104 | 0.013 |
| IVKF-C ($W = P^{-1}$) | 2.842 | 0.204 | 0.097 | 0.016 |

The RMSE performance of PL-MMSE-C ($W = I$) is $\text{RMSE}_{avg}^{pos} = 1.781$ m and $\text{RMSE}_{avg}^{vel} = 0.103$ m/s at large bearing noise level $\sigma_\theta = 7°$, which are less than other constrained algorithms. Similarly, the BNorm performance of PL-MMSE-C ($W = I$) has $\text{BNorm}_{avg}^{pos} = 0.084$ m and $\text{BNorm}_{avg}^{vel} = 0.001$ m/s at large bearing noise level $\sigma_\theta = 7°$ which are better than the others. The numerical results of PL-MMSE-C ($W = P^{-1}$) are similar to that of PL-MMSE-C ($W = I$), as observed from the table. The filters combined with constraints can achieve better performance from Figures 6 and 7 and Table 5. As shown in Table 5, the constraint algorithms with $W = P^{-1}$ are not necessarily better than the corresponding filters with $W = I$, where $\text{RMSE}_{avg}^{pos} = 2.537$ m of IVKF-C ($W = I$) is less than $\text{RMSE}_{avg}^{pos} = 2.842$ m of IVKF-C ($W = P^{-1}$) and $\text{RMSE}_{avg}^{pos} = 2.890$ m of BC-PLKF-C ($W = I$) is greater than $\text{RMSE}_{avg}^{pos} = 2.883$ m of BC-PLKF-C ($W = P^{-1}$). This difference is caused by the discrepancy between the actual distribution characteristics of the state estimate $\hat{x}$ and $P$. When the distribution is close to $P$, the constraint algorithms with $W = P^{-1}$ behave better than the corresponding constraint algorithms with $W = I$.

In the second scenario, the target moves on an arc, as shown in Figure 3b, with a nearly constant velocity magnitude $V$. The turning center is $(R_x, R_y)$ with the radius $R$. Hence, the constrained equation $h(\cdot)$, the vector $g$, the constrained matrix $M$, the vector $m$ and the variable $m_0$, are given by

$$h(x) = (x(1) - R_x)^2 + (x(2) - R_y)^2, \tag{76}$$

$$g = R^2, \tag{77}$$

$$M = \begin{bmatrix} 1 & 0 \\ 0 & 1 \end{bmatrix}, \tag{78}$$

$$m = -\begin{bmatrix} R_x & R_y \end{bmatrix}^T, \tag{79}$$

$$m_0 = R_x^2 + R_y^2 - R^2. \tag{80}$$

In addition, the velocity constraint is introduced into the state estimation. The constrained velocity estimate $\tilde{v}$ is

$$\tilde{v} = \left( \hat{v}^T \mu \right) \mu \tag{81}$$

where the unconstrained velocity estimate $\hat{v}$ and constrained unit direction vector $\mu$ are

$$\hat{v} = [\hat{x}(3)\ \hat{x}(4)]^T, \tag{82}$$

$$\mu = [-\sin\theta\ \cos\theta]^T \tag{83}$$

with $\theta = \tan^{-1}(\hat{x}(2)/\hat{x}(1))$.

In the simulation, the sampling interval $T$ is set to 0.1 s. The total time scan is the 20 s. The turning center $(R_x, R_y)$ is set to $(100, 0)$ m. The turning radius $R$ is 100 m. The initial position for the target is $[200\ 0]^T$ m. To compare the linear approximation with the second-order approximation method, two experiments are carried out, where the magnitude of $V$ is $10^{-4}$ m/s and 0.2 m/s, respectively.

For $V = 10^{-4}$ m/s, the true initial state is $x_1 = \begin{bmatrix} 200\ 0\ 0\ 10^{-4} \end{bmatrix}^T$. The initial covariance matrix is $P_{1|1} = \text{diag}([10^{-8}\ 10^{-8}\ 10^{-10}\ 10^{-10}])$. The process noise $\omega_{k-1}$ and bearing noise $e_k$ have the same composition as (74) and (75) where $\mu_{x1}^T = \begin{bmatrix} 0 & 0 & 0 & 0 \end{bmatrix}$, $\mu_{x2}^T = \begin{bmatrix} 0 & 0 & 0 & 0 \end{bmatrix}$, $\mu_{z1} = 0$, $\mu_{z2} = 0$, $Q_1 = \text{diag}[0\ 0\ 10^{-9}\ 10^{-9}]$, $Q_2 = \text{diag}([0\ 0\ 2 \times 10^{-9}\ 2 \times 10^{-9}])$, $R_1 = 1.5 \times 10^{-12}$, $R_2 = 1.909 \times 10^{-12}$ and $\lambda = 0.4$. The standard deviation $\sigma_\theta$ of bearing noise for the corresponding $\rho$ is set as shown in Table 6.

**Table 6.** Standard deviation $\sigma_\theta$ against $\rho$ at $V = 10^{-4}$ m/s.

| $\rho$ | 1 | 2 | 3 | 4 | 5 | 6 | 7 | 8 | 9 | 10 |
|---|---|---|---|---|---|---|---|---|---|---|
| $\sigma_\theta(\times 10^{-5}°)$ | 1 | 2 | 3 | 4 | 5 | 6 | 7 | 8 | 9 | 10 |

Simulation results of the time-averaged RMSEs and BNorms of the target position and velocity estimates and the bearing noise standard deviations is shown in Figure 8 where PL-MMSE has lower errors in the RMSE performance with $\text{RMSE}_{avg}^{pos} = 2.526 \times 10^{-4}$ m, $\text{RMSE}_{avg}^{vel} = 1.783 \times 10^{-5}$ m/s.

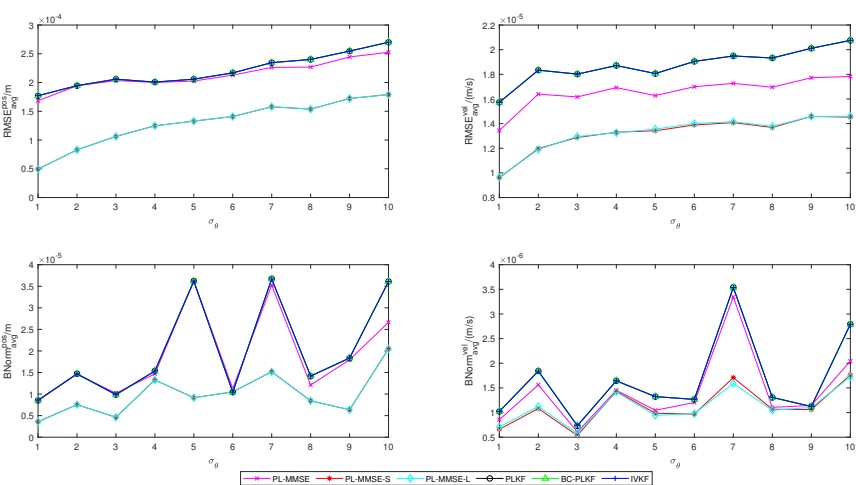

**Figure 8.** Time-averaged RMSEs, BNorms and bearing noise standard deviation at $V = 10^{-4}$ m/s for the PL-MMSE, PLKF, BC-PLKF and IVKF algorithms, as well as, the PL-MMSE algorithm with nonlinear constraints proposed in the paper.

The evolution of RMSEs, BNorms of the target position and velocity estimates in time $kT$ for $\sigma_\theta = 7°$ at $V = 10^{-4}$ m/s is presented in Figure 9.

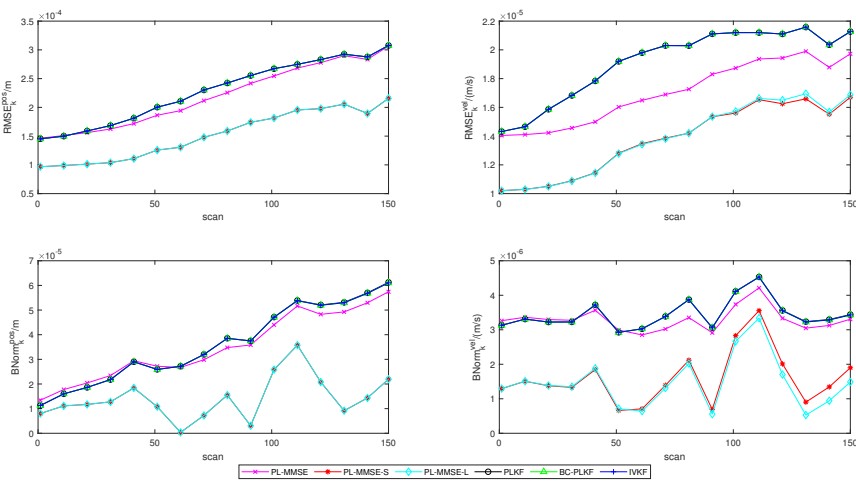

**Figure 9.** RMSEs, BNorms and time $kT$ for $\sigma_\theta = 7°$ at $V = 10^{-4}$ m/s for the PL-MMSE, PLKF, BC-PLKF and IVKF algorithms, as well as, the proposed PL-MMSE algorithm with nonlinear constraints in the paper.

It is noticeable that the curve trends in Figures 8 and 9 are similar to Figures 4 and 5 because of the weak model nonlinearity caused by small velocity. The performance metric values of PL-MMSE gradually approach $\text{RMSE}_k^{pos} = 3.099 \times 10^{-4}$ m, $\text{RMSE}_k^{vel} = 2.053 \times 10^{-5}$ m/s, $\text{BNorm}_k^{pos} = 5.774 \times 10^{-5}$ m and $\text{BNorm}_k^{vel} = 3.497 \times 10^{-6}$ m/s, which are superior to other unconstrained algorithms in Figure 9. It is also demonstrated that the constrained PL-MMSE is better than the unconstrained PL-MMSE and other constrained filters at all bearing noise levels under the arc section in Figures 8 and 9. Performance comparisons of four algorithms combined with the linear approximation method and the second-order

approximation method for $\sigma_\theta = 7°$ at $V = 10^{-4}$ m/s are shown in Figures 10 and 11, respectively, which present that PL-MMSE-L and PL-MMSE-S have fewer errors than other constrained algorithms at a large bearing noise level.

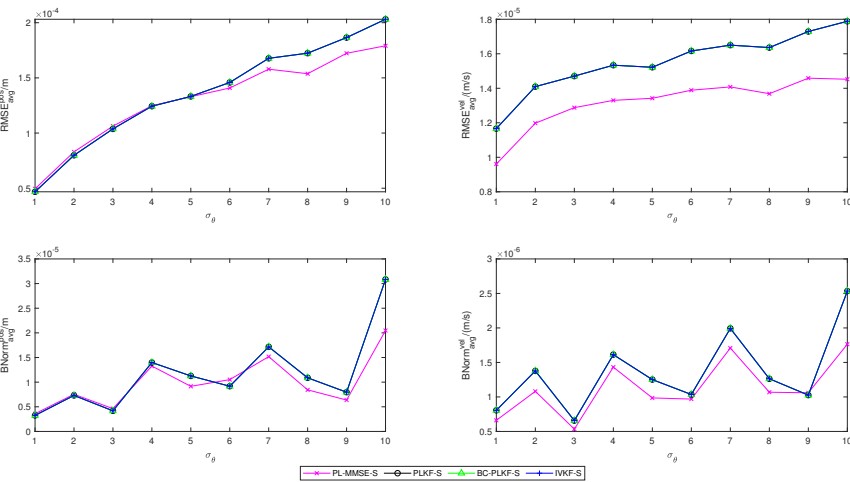

**Figure 10.** Time-averaged RMSEs, BNorms and bearing noise standard deviation at $V = 10^{-4}$ m/s for the four algorithms combined with the linear approximation method.

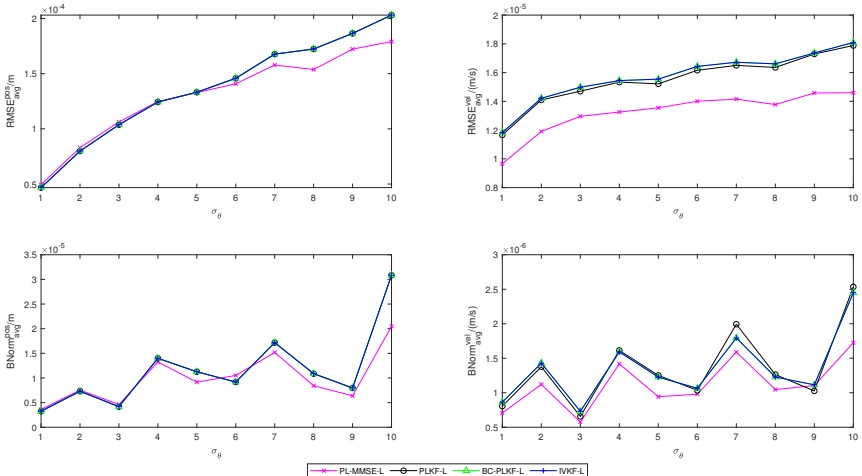

**Figure 11.** Time-averaged RMSEs, BNorms and bearing noise standard deviation at $V = 10^{-4}$ m/s for the four algorithms combined with the second-order approximation method.

Table 7 presents the results of the time-averaged RMSEs and BNorms of different filters for $\sigma_\theta = 7°$ at $V = 10^{-4}$ m/s. The RMSE and BNorm performance of PL-MMSE-L are $\text{RMSE}_{avg}^{pos} = 1.579 \times 10^{-4}$ m, $\text{RMSE}_{avg}^{vel} = 1.415 \times 10^{-5}$ m/s, $\text{BNorm}_{avg}^{pos} = 1.518 \times 10^{-5}$ m and $\text{BNorm}_{avg}^{vel} = 1.590 \times 10^{-6}$ m/s. The PL-MMSE algorithm combined with the linear approximation method is better than the second-order approximation method as $\text{BNorm}_{avg}^{vel} = 1.708 \times 10^{-6}$ m/s of the PL-MMSE-S is greater than $\text{BNorm}_{avg}^{vel} = 1.590 \times 10^{-6}$ m/s of the PL-MMSE-L in Table 7 due to the weak nonlinearity.

For $V = 0.2$ m/s, we have the true initial state $x_1 = [200\ 0\ 0\ 0.2]^T$ and the initial covariance matrix $P_{1|1} = \text{diag}([10^{-3}\ 10^{-3}\ 10^{-4}\ 10^{-4}])$. The composition of the process noise $\omega_{k-1}$, and the bearing noise $e_k$ is the same as (74) and (75), where $\mu_{x1}^T = [0\ 0\ 0\ 0]$, $\mu_{x2}^T = [0\ 0\ 0\ 0]$, $\mu_{z1} = 0$, $\mu_{z2} = 0$, $Q_1 = \text{diag}([0\ 0\ 0.01\ 0.01])$, $Q_2 = \text{diag}([0\ 0\ 0.02\ 0.02])$, $R_1 = 1.5 \times 10^{-3}$, $R_2 = 1.909 \times 10^{-3}$ and $\lambda = 0.4$. Table 8 presents the standard deviation $\sigma_\theta$ of the bearing noise for the corresponding $\rho$.

**Table 7.** $\text{RMSE}_{avg}^{pos}$ ($\times 10^{-4}$ m), $\text{RMSE}_{avg}^{vel}$ ($\times 10^{-5}$ m), $\text{BNorm}_{avg}^{pos}$ ($\times 10^{-5}$ m) and $\text{BNorm}_{avg}^{vel}$ ($\times 10^{-6}$ m) of different filters for $\sigma_\theta = 7°$ at $V = 10^{-4}$ m/s on the arc section.

| Filter | $\text{RMSE}_{avg}^{pos}$ | $\text{RMSE}_{avg}^{vel}$ | $\text{BNorm}_{avg}^{pos}$ | $\text{BNorm}_{avg}^{vel}$ |
|---|---|---|---|---|
| PL-MMSE | 2.263 | 1.726 | 3.528 | 3.344 |
| PLKF | 2.346 | 1.949 | 3.673 | 3.541 |
| BC-PLKF | 2.345 | 1.949 | 3.667 | 3.537 |
| IVKF | 2.345 | 1.949 | 3.667 | 3.537 |
| PL-MMSE-L | 1.579 | 1.415 | 1.518 | 1.590 |
| PL-MMSE-S | 1.579 | 1.408 | 1.518 | 1.708 |
| PLKF-L | 1.677 | 1.650 | 1.716 | 1.994 |
| PLKF-S | 1.677 | 1.650 | 1.716 | 1.994 |
| BC-PLKF-L | 1.677 | 1.672 | 1.716 | 1.794 |
| BC-PLKF-S | 1.677 | 1.650 | 1.716 | 1.993 |
| IVKF-L | 1.677 | 1.672 | 1.716 | 1.794 |
| IVKF-S | 1.677 | 1.650 | 1.716 | 1.993 |

**Table 8.** Standard deviation $\sigma_\theta$ against $\rho$ at $V = 0.2$ m/s.

| $\rho$ | 1 | 2 | 3 | 4 | 5 | 6 | 7 | 8 | 9 | 10 |
|---|---|---|---|---|---|---|---|---|---|---|
| $\sigma_\theta (/\sqrt{10}°)$ | 1 | 2 | 3 | 4 | 5 | 6 | 7 | 8 | 9 | 10 |

Simulation results of the time-averaged RMSEs and BNorms of the target position and velocity estimates against $\sigma_\theta$ are presented in Figure 12.

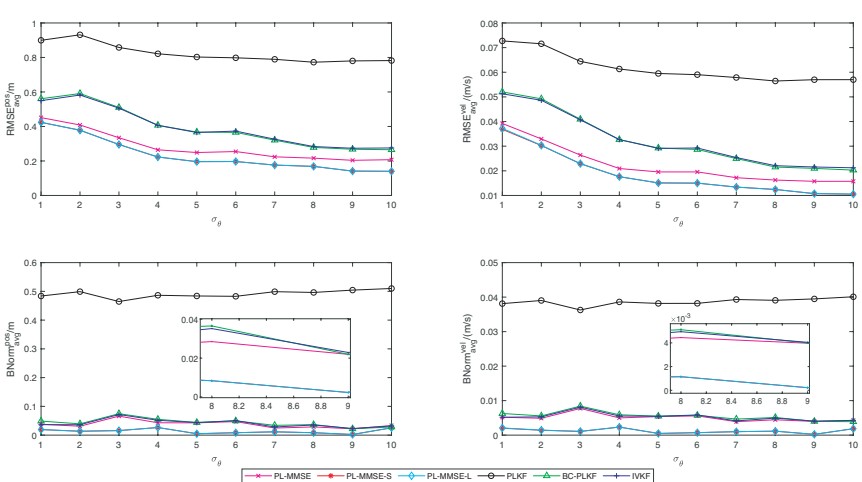

**Figure 12.** Time-averaged RMSEs, BNorms and bearing noise standard deviation at $V = 0.2$ m/s for the PL-MMSE, PLKF, BC-PLKF and IVKF algorithms, as well as, the PL-MMSE algorithm with nonlinear constraints proposed in the paper.

The performance of PL-MMSE and PL-MMSE with constraints is better than other filters, where $\text{RMSE}_{avg}^{pos} = 0.3346$ m of PL-MMSE and $\text{RMSE}_{avg}^{pos} = 0.2954$ m of PL-MMSE-S are less than other algorithms for $\sigma_\theta = 3°$. The evolution of RMSEs, BNorms of the target position and velocity estimates in time $kT$ for $\sigma_\theta = 7°$ at $V = 0.2$ m/s is shown in Figure 13.

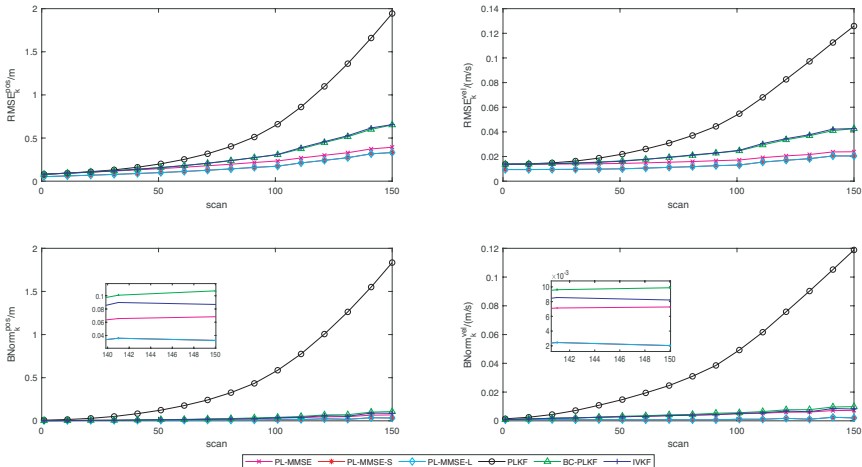

**Figure 13.** RMSEs, BNorms and time $kT$ for $\sigma_\theta = 7°$ at $V = 0.2$ m/s for the PL-MMSE, PLKF, BC-PLKF and IVKF algorithms, as well as, the proposed PL-MMSE algorithm with nonlinear constraints in the paper.

It is evident that the constrained PL-MMSE has more stable and accurate tracking performance at large bearing noise levels under the arc section. Comparisons of four algorithms combined with the linear approximation method and the second-order approximation method for $\sigma_\theta = 7°$ at $V = 0.2$ m/s are provided in Figures 14 and 15, respectively. The time-averaged RMSEs and BNorms of different filters for $\sigma_\theta = 7°$ are presented in Table 9. It is remarkable that the PL-MMSE with constraints has less errors than other filters, where $\text{RMSE}_{avg}^{pos}$ of PL-MMSE-L is 0.176 m and $\text{RMSE}_{avg}^{pos}$ of PL-MMSE-S is 0.176 m. On contrary to the first experiment with $V = 10^{-4}$ m/s, the algorithms combined with the second-order approximation method perform better than the algorithms combined with the linear approximation method in Table 9, where $\text{RMSE}_{avg}^{pos}$ of PL-MMSE-S is less than $\text{RMSE}_{avg}^{pos}$ of PL-MMSE-L for $\sigma_\theta = 7°$ at the scale of $10^{-6}$ because of strong nonlinearity.

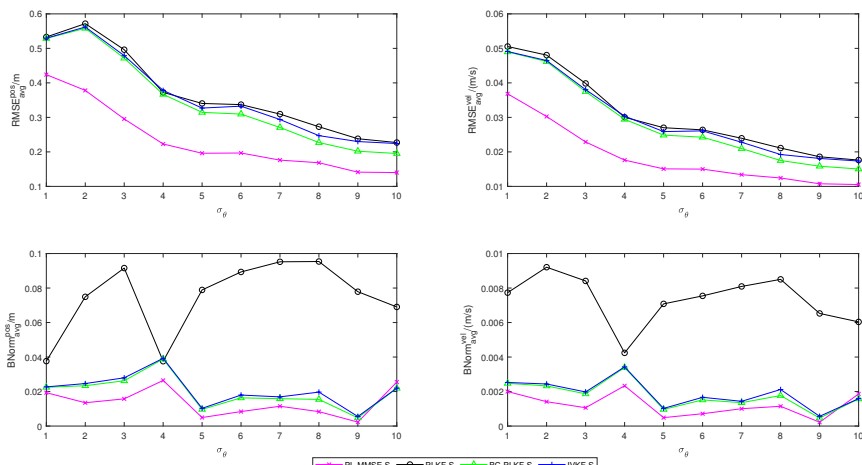

**Figure 14.** Time-averaged RMSEs, BNorms and bearing noise standard deviation at $V = 0.2$ m/s for the four algorithms combined with the linear approximation method.

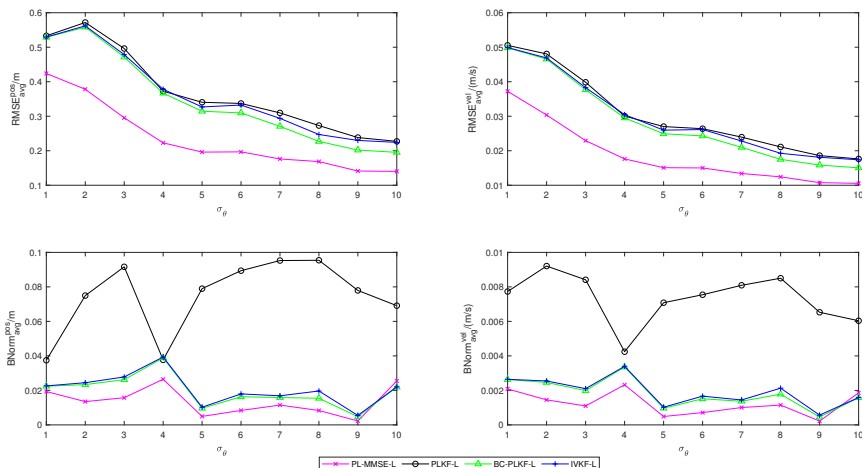

**Figure 15.** Time-averaged RMSEs, BNorms and bearing noise standard deviation at $V = 0.2$ m/s for the four algorithms combined with the second-order approximation method.

**Table 9.** $\text{RMSE}_{avg}^{pos}$, $\text{RMSE}_{avg}^{vel}$, $\text{BNorm}_{avg}^{pos}$ and $\text{BNorm}_{avg}^{vel}$ of different filters for $\sigma_\theta = 7°$ at $V = 0.2$ m/s on the arc section.

| Filter | $\text{RMSE}_{avg}^{pos}$ | $\text{RMSE}_{avg}^{vel}$ | $\text{BNorm}_{avg}^{pos}$ | $\text{BNorm}_{avg}^{vel}$ |
|---|---|---|---|---|
| PL-MMSE | 0.224 | 0.017 | 0.025 | 0.004 |
| PLKF | 0.789 | 0.058 | 0.499 | 0.039 |
| BC-PLKF | 0.321 | 0.025 | 0.034 | 0.005 |
| IVKF | 0.326 | 0.025 | 0.028 | 0.004 |
| PL-MMSE-L | 0.176 | 0.013 | 0.011 | 0.001 |
| PL-MMSE-S | 0.176 | 0.013 | 0.011 | 0.001 |
| PLKF-L | 0.310 | 0.024 | 0.095 | 0.008 |
| PLKF-S | 0.310 | 0.024 | 0.095 | 0.008 |
| BC-PLKF-L | 0.271 | 0.021 | 0.016 | 0.001 |
| BC-PLKF-S | 0.271 | 0.021 | 0.016 | 0.001 |
| IVKF-L | 0.294 | 0.023 | 0.017 | 0.001 |
| IVKF-S | 0.294 | 0.023 | 0.017 | 0.001 |

It is worthwhile to point out that the errors of the filters are basically unchanged and slightly decrease as the noise level rises in Figure 12 when the velocity is relatively large. The cause of this phenomenon is the error in the state update equation introduced by the linearization of the arc movement. When $V = 0.2$ m/s, the error from the state update equation becomes the main source of the filter estimate error, which relatively reduces the effect of the bearing noise and breaks the average algorithm performance trend.

**Remark 2.** *It is noticeable that the magnitudes of the velocity in the second scenario are both small. There are two reasons for such a setting. Firstly, the PL-MMSE is still a linear Kalman filter under linear and nonlinear constraints. There is discrepancy between the TMA result from the linear motion model (9) and the actual arc trajectory in the circular section. The faster the target moves in an interval, the greater the discrepancy will be. Secondly, the sampling frequency of the sensor in reality is much higher than that in our experiment. For example, the sampling frequency of the radar is generally between 1 and 15 GHz. When the sampling frequency increases, the relative speed of the target also rises up proportionally to maintain the same traveling distance. Hence, if we set the sampling time $T = 10^{-5}$ s rather than 0.1 s as in the simulation, the relative speed of the target becomes $V = 1$ m/s and 2000 m/s, respectively, which are quite common in practice. The transformation of the sampling frequency and target velocity corresponding to a real radar demonstrates that the setting is meaningful.*

*Since the distance the target moves in an interval is small, the corresponding RMSEs are low. Nevertheless, simulation results show that the RMSEs of our constrained algorithms are much smaller than those of other filters.*

## 6. Conclusions and Future Works

In this paper, we propose a new pseudolinear Kalman filtering method based on TMA with available state constraints by combining the PL-MMSE and state constraints. The mean square and estimation projection methods are enveloped with PL-MMSE to address the linear constrained state estimation problem. The linear approximation and second-order approximation methods are used to refine PL-MMSE estimates under nonlinear constraints. The merged algorithms can effectively solve the bearings-only TMA problem under Gaussian mixture noise. Simulations show that the constrained PL-MMSE has better performance than other filters. In particular, when the target velocity is small, the algorithms combined with the linear approximation method perform better than those combined with the second-order approximation method under the circular road. It turns out just the opposite when the velocity is large.

Analyzing the statistical properties of the PL-MMSE with state constraints will be one topic of our future research. Applying the designed algorithms in actual engineering practice is the other direction we endeavor to study for the next step.

## 7. Patents

Yiqun Zou, Shuang Zou. A target tracking and positioning method, system, device and readable storage medium: ZL202110038824.8[P].2021.1.12.

**Author Contributions:** Conceptualization, M.L.; methodology, M.L.; software, M.L.; validation, Y.Z., M.L. and X.T.; formal analysis, Y.Z.; investigation, M.L.; resources, Y.Z.; simulation, M.L.; writing—original draft preparation, M.L.; writing—review and editing, M.L. and Y.Z.; visualization, X.T.; supervision, Q.Z.; project administration, X.T.; funding acquisition, Y.Z. All authors have read and agreed to the published version of the manuscript.

**Funding:** This work is supported by National Natural Science Foundation of China (NSFC) [grant 61403427] and Hunan Provincial Natural Science Foundation of China [project 2020JJ5585 and project 2020JJ5777].

**Institutional Review Board Statement:** Not applicable.

**Informed Consent Statement:** Not applicable.

**Data Availability Statement:** Not applicable.

**Acknowledgments:** The authors would like to thank Shuang Zou for his valuable suggestions on this paper.

**Conflicts of Interest:** The authors declare no conflict of interest. The funders had no role in the design of the study; in the collection, analyses, or interpretation of data; in the writing of the manuscript, or in the decision to publish the results.

## Abbreviations

The following abbreviations are used in this manuscript:

| | |
|---|---|
| TMA | Target motion analysis |
| PLKF | Pseudolinear Kalman filter |
| PL-MMSE | Pseudolinear Kalman filter under the minimum mean square error framework |
| AOA | Angle of arrival |
| BC-PLKF | Bias-compensated PLKF |
| IVKF | IV Kalman filter |
| SAM-IVKF | IVKF based on selective-angle-measurement |
| 2D | Two-dimensional |
| RMSEs | Root mean square errors |
| BNorms | Bias norms |

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
