# Peer review of "Non-Gaussian Pseudolinear Kalman Filtering-Based Target Motion Analysis with State Constraints"

_applsci, doi:10.3390/app12199975_

Round 1
Reviewer 1 Report
As can be seen in Figure 3, in this paper, the movement of the target is set as the driving of the vehicle.
In addition, in the second scenario, the target moves on an arc, as shown in Figure 3(b) that a land vehicle travels along a circular road segment with nearly constant velocity magnitude V.
By the way, two experiments are carried out where the magnitude of V is 10^{-4}m/s and 0.2m/s, respectively.
Both velocities are so small. Especially the assumption of 10^{-4}m/s is the level at which the car is almost stationary.
As a result, RMSEs in Figure 8 ~ Figure 11 are too small. I'm not sure what this situation means.
Author Response
Please check the letter attached. Thank you!

Reviewer 2 Report
To make a decision about the possibility of publishing an article, the authors need to answer the following questions:
- where in reality the simplest object model considered in the article can be used,
- why the change of sensor coordinates is not taken into account in the model,
- what is the point of using Gaussian mixture noise instead of standard Gaussian noise with relevant characteristics (sum of Gaussian noises is Gaussian noise).
Author Response

(The authors gave the same response as above.)

Round 2
Reviewer 1 Report
All authors' responses are satisfactory.